# BIVA: A Very Deep Hierarchy of Latent Variables for Generative Modeling

**Lars Maaløe**
Corti
Copenhagen
Denmark
lm@corti.ai

**Marco Fraccaro**
Unumed
Copenhagen
Denmark
mf@unumed.com

**Valentin Liévin & Ole Winther**
Technical University of Denmark
Copenhagen
Denmark
{valv,olwi}@dtu.dk

## Abstract

With the introduction of the variational autoencoder (VAE), probabilistic latent variable models have received renewed attention as powerful generative models. However, their performance in terms of test likelihood and quality of generated samples has been surpassed by autoregressive models without stochastic units. Furthermore, flow-based models have recently been shown to be an attractive alternative that scales well to high-dimensional data. In this paper we close the performance gap by constructing VAE models that can effectively utilize a deep hierarchy of stochastic variables and model complex covariance structures. We introduce the Bidirectional-Inference Variational Autoencoder (BIVA), characterized by a skip-connected generative model and an inference network formed by a bidirectional stochastic inference path. We show that BIVA reaches state-of-the-art test likelihoods, generates sharp and coherent natural images, and uses the hierarchy of latent variables to capture different aspects of the data distribution. We observe that BIVA, in contrast to recent results, can be used for anomaly detection. We attribute this to the hierarchy of latent variables which is able to extract high-level semantic features. Finally, we extend BIVA to semi-supervised classification tasks and show that it performs comparably to state-of-the-art results by generative adversarial networks.

## 1 Introduction

One of the key aspirations in recent machine learning research is to build models that *understand the world* [24, 40, 11, 57]. Generative models are providing the means to learn from a plethora of unlabeled data in order to model a complex data distribution, e.g. natural images, text, and audio. These models are evaluated by their ability to *generate* data that is similar to the input data distribution from which they were trained on. The range of applications that come with generative models are vast, where audio synthesis [55] and semi-supervised classification [38, 31, 44] are examples hereof. Generative models can be broadly divided into explicit and implicit density models. The generative adversarial network (GAN) [11] is an example of an implicit model, since it is not possible to procure a likelihood estimation from this model framework. The focus of this research is instead within explicit density models, for which a tractable or approximate likelihood estimation can be performed.

The three main classes of powerful explicit density models are autoregressive models [26, 57], flow-based models [8, 9, 21, 16], and probabilistic latent variable models [24, 40, 33]. In recent years autoregressive models, such as the PixelRNN and the PixelCNN [57, 45], have achieved superior likelihood performance and flow-based models have proven efficacy on large-scale natural image generation tasks [21]. However, in the autoregressive models, the runtime performance of generation is scaling poorly with the complexity of the input distribution. The flow-based models do not possess

this restriction and do indeed generate visually compelling natural images when sampling close to the mode of the distribution. However, generation from the actual learned distribution is still not outperforming autoregressive models [21, 16].

Probabilistic latent variable models such as the variational auto-encoder (VAE) [24, 40] possess intriguing properties that are different from the other classes of explicit density models. They are characterized by a posterior distribution over the latent variables of the model, derived from Bayes' theorem, which is typically intractable and needs to be approximated. This distribution most commonly lies on a low-dimensional manifold that can provide insights into the internal representation of the data [1]. However, the latent variable models have largely been disregarded as powerful generative models due to *blurry* generations and poor likelihood performances on natural image tasks. [27, 10], amongst others, attribute this tendency to the usage of a similarity metric in pixel space. Contrarily, we attribute it to the lack of overall model expressiveness for accurately modeling complex input distributions, as discussed in [59, 41].

There has been much research into explicitly defining and learning more expressive latent variable models. Here, the complementary research into learning a covariance structure through a framework of normalizing flows [39, 52, 23] and the stacking of a hierarchy of latent variables [4, 37, 31, 50] have shown promising results. However, despite significant improvements, the reported performance of these models has still been inferior to their autoregressive counterparts. This has spawned a new class of explicit density models that adds an autoregressive component to the generative process of a latent variable model [14, 5]. In this combination of model paradigms, the latent variables can be viewed as merely a *lossy* representation of the input data and the model still suffers from the same issues as autoregressive models.

**Contributions.**     In this research we argue that latent variable models that are defined in a sufficiently expressive way can compete with autoregressive and flow-based models in terms of test log-likelihood and quality of the generated samples. We introduce the Bidirectional-Inference Variational Autoencoder (BIVA), a model formed by a deep hierarchy of stochastic variables that uses skip-connections to enhance the flow of information and avoid inactive units. To define a flexible posterior approximation, we construct a bidirectional inference network using stochastic variables in a bottom-up and a top-down inference path. The inference model is reminiscent to the stochastic top-down path introduced in the Ladder VAE [50] and IAF VAE [50] with the addition that the bottom-up pass is now also stochastic and there are no autoregressive components. We perform an in-depth analysis of BIVA and show **(i)** an ablation study that analyses the contributions of the individual novel components, **(ii)** that the model is able to improve on state-of-the-art results on benchmark image datasets, **(iii)** that a small extension of the model can be used for semi-supervised classification and performs comparably to current state-of-the-art models, and **(iv)** that the model, contrarily to other state-of-the-art explicit density models [34], can be utilized for anomaly detection on complex data distributions.

## 2   Variational Autoencoders

The VAE is a generative model parameterized by a neural network $\theta$ and is defined by an observed variable $x$ that depends on a hierarchy of stochastic latent variables $\mathbf{z} = z_1, ..., z_L$ so that: $p_\theta(x, \mathbf{z}) = p_\theta(x|z_1)p_\theta(z_L)\prod_{i=1}^{L-1} p_\theta(z_i|z_{i+1})$. The posterior distribution over the latent variables of a VAE is commonly analytically intractable, and is approximated with a variational distribution which is factorized with a bottom-up structure, $q_\phi(\mathbf{z}|x) = q_\phi(z_1|x)\prod_{i=1}^{L-1} q_\phi(z_{i+1}|z_i)$, so that each latent variable is conditioned on the variable below in the hierarchy. The parameters $\theta$ and $\phi$ can be optimized by maximizing the *evidence lower bound* (ELBO)

$$\log p_\theta(x) \geq \mathbb{E}_{q_\phi(\mathbf{z}|x)}\left[\log \frac{p_\theta(x, \mathbf{z})}{q_\phi(\mathbf{z}|x)}\right] \equiv \mathcal{L}(\theta, \phi)\,. \tag{1}$$

A detailed introduction on VAEs can be found in appendix A in the supplementary material. While a deep hierarchy of latent stochastic variables will result in a more expressive model, in practice the top stochastic latent variables of standard VAEs have a tendency to *collapse* into the prior. The Ladder VAE (LVAE) [50] is amongst the first attempts towards VAEs that can effectively leverage multiple layers of stochastic variables. This is achieved by parameterizing the variational approximation with a *bottom-up* deterministic path followed by a *top-down* inference path that shares parameters with

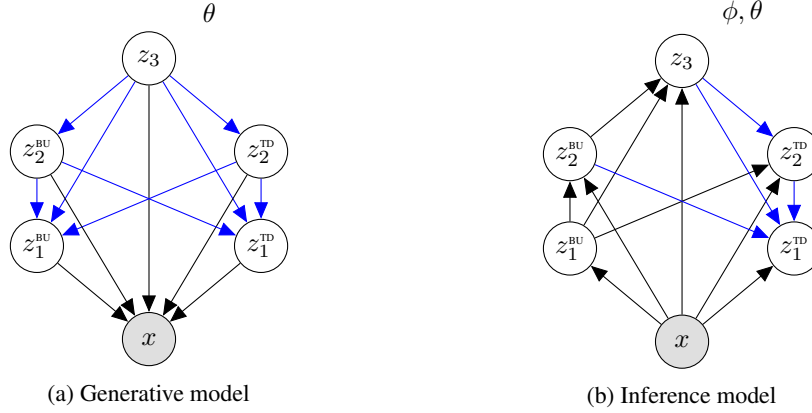

| (a) Generative model | (b) Inference model |

Figure 1: A $L = 3$ layered BIVA with (a) the generative model and (b) inference model. Blue arrows indicate that the deterministic parameters are shared between the inference and generative models. See Appendix B for a detailed explanation and a graphical model that includes the deterministic variables.

the top-down structure of the generative model: $q_{\phi,\theta}(\mathbf{z}|x) = q_\phi(z_L|x) \prod_{i=1}^{L-1} q_{\phi,\theta}(z_i|z_{i+1}, x)$. See Appendix A for a graphical representation of the LVAE inference network. Thanks to the bottom-up path, all the latent variables in the hierarchy have a deterministic dependency on the observed variable $x$, which allows data-dependent information to skip all the stochastic variables lower in the hierarchy (Figure 5d in Appendix A). The stochastic latent variables that are higher in the hierarchy will therefore receive less noisy inputs, and will be empirically less likely to collapse. Despite the improvements obtained thanks to the more flexible inference network, in practice LVAEs with a very deep hierarchy of stochastic latent variables will still experience variable collapse. In the next section we will introduce the Bidirectional-Inference Variational Autoencoder, that manages to avoid these issues by extending the LVAE in 2 ways: (i) adding a deterministic top-down path in the generative model and (ii) defining a factorization of the latent variables $z_i$ at each level of the hierarchy that allows to construct a bottom-up *stochastic* inference path.

## 3    Bidirectional-Inference Variational Autoencoder

In this section, we will first describe the architecture of the Bidirectional-Inference Variational Autoencoder (Figure 1), and then provide the motivation behind the main ideas of the model as well as some intuitions on the role of each of its novel components. Finally, we will show how this model can be used for a novel approach to detecting anomalous data.

### 3.1    Model architecture

**Generative model.**    In BIVA, at each layer $1, ..., L - 1$ of the hierarchy we split the latent variable in two components, $z_i = (z_i^{\text{BU}}, z_i^{\text{TD}})$, which belong to a bottom-up (BU) and top-down (TD) inference path, respectively. More details on this will be given when introducing the inference network. The generative model of BIVA is illustrated in Figure 1a. We introduce a deterministic top-down path $d_{L-1}, \ldots, d_1$ that is parameterized with neural networks and receives as input at each layer $i$ of the hierarchy the latent variable $z_{i+1}$. In the case of a convolutional model, this is done by concatenating $(z_{i+1}^{\text{BU}}, z_{i+1}^{\text{TD}})$ and $d_{i+1}$ along the features' dimension. $d_i$ can therefore be seen as a deterministic variable that summarizes all the relevant information coming from the stochastic variables higher in the hierarchy, $z_{>i}$. The latent variables $z_i^{\text{BU}}$ and $z_i^{\text{TD}}$ are conditioned on all the information in the higher layers, and are conditionally independent given $z_{>i}$. The joint distribution of the model is then given by:

$$p_\theta(x, \mathbf{z}) = p_\theta(x|\mathbf{z})p_\theta(z_L) \prod_{i=1}^{L-1} p_\theta(z_i^{\text{BU}}|z_{>i})p_\theta(z_i^{\text{TD}}|z_{>i}) \,,$$

where $\theta$ are the parameters of the generative model. The likelihood of the model $p_\theta(x|\mathbf{z})$ directly depends on $z_1$, and depends on $z_{>1}$ through the deterministic top-down path. Each stochastic latent

variable $1, ..., L$ is parameterized by a Gaussian distribution with diagonal covariance, with one neural network $\mu(\cdot)$ for the mean and another neural network $\sigma(\cdot)$ for the variance. Since the $z_{i+1}^{\text{BU}}$ and $z_{i+1}^{\text{TD}}$ variables are on the same level in the generative model and of the same dimensionality, we share all the deterministic parameters going to the layer below. See Appendix B for details.

**Bidirectional inference network.**  Due to the non-linearities in the neural networks that parameterize the generative model, the exact posterior distribution $p_\theta(\mathbf{z}|x)$ is intractable and needs to be approximated. As for VAEs, we therefore define a variational distribution, $q_\phi(\mathbf{z}|x)$, that needs to be flexible enough to approximate the true posterior distribution, as closely as possible. We define a bottom-up (BU) and a top-down (TD) inference path, which are computed sequentially when constructing the posterior approximation for each data point $x$, see Figure 1b. The variational distribution over the BU latent variables depends on the data $x$ and on all BU variables lower in the hierarchy, i.e. $q_\phi(z_i^{\text{BU}}|x, z_{<i}^{\text{BU}})$, where $\phi$ denotes all the parameters of the BU path. $z_i^{\text{BU}}$ has a direct dependency only on the BU variable below, $z_{i-1}^{\text{BU}}$. The dependency on $z_{<i-1}^{\text{BU}}$ is achieved, similarly to the generative model, through a deterministic bottom-up path $\widetilde{d}_1, \ldots, \widetilde{d}_{L-1}$.

The TD variables depend on the data and the BU variables lower in the hierarchy through the BU inference path, but also on all variables above in the hierarchy through the TD inference path, see Figure 1b. The variational approximation over the TD variables is thereby $q_{\phi,\theta}(z_i^{\text{TD}}|x, z_{<i}^{\text{BU}}, z_{>i}^{\text{BU}}, z_{>i}^{\text{TD}})$. Importantly, all the parameters of the TD path are shared with the generative model, and are therefore denoted as $\theta$. The overall inference network can be factorized as follows:

$$q_\phi(\mathbf{z}|x) = q_\phi(z_L|x, z_{<L}^{\text{BU}}) \prod_{i=1}^{L-1} q_\phi(z_i^{\text{BU}}|x, z_{<i}^{\text{BU}}) q_{\phi,\theta}(z_i^{\text{TD}}|x, z_{<i}^{\text{BU}}, z_{>i}^{\text{BU}}, z_{>i}^{\text{TD}}) \,,$$

where the variational distributions over the BU and TD latent variables are Gaussians whose mean and diagonal covariance are parameterized with neural networks that take as input the concatenation over the feature dimension of the conditioning variables. Training of BIVA is performed, as for VAEs, by maximizing the ELBO in eq. (1) with stochastic backpropagation and the reparameterization trick.

## 3.2   Motivation

BIVA can be seen as an extension of the LVAE in which we (i) add a deterministic top-down path and (ii) apply a bidirectional inference network. We will now provide the motivation and some intuitions on the role of these two novel components, that will then be empirically validated with the ablation study of Section 4.1.

**Deterministic top-down path.**  Skip-connections represent one of the simplest yet most powerful advancements of deep learning in recent years. They allow constructing very deep neural networks, by better propagating the information throughout the model and reducing the issue of vanishing gradients. Skip connections form for example the backbone of deep neural networks such as ResNets [15], which have shown impressive performances on a wide range of classification tasks. Our goal in this paper is to build very deep latent variable models that are able to learn an expressive latent hierarchical representation of the data. In our experiments, we however found that the LVAE still had difficulties in activating the top latent variables for deeper hierarchies. To limit this issue, we add skip connections among the latent variables in the generative model by adding the deterministic top-down path, that makes each variable depend on all the variables above in the hierarchy (see Figure 1a for a graphical representation). This allows a better flow of information in the model and thereby avoids the collapse of latent variables. A related idea was recently proposed by [7], that add skip connections among the neural network layers parameterizing a shallow VAE with a single latent variable.

**Bidirectional inference.**  The inspiration for the bidirectional inference network of BIVA comes from the work on Auxiliary VAEs (AVAE) by [37, 31]. An AVAE can be viewed as a shallow VAE with a single latent variable $z$ and an auxiliary variable $a$ that increases the expressiveness of the variational approximation $q_\phi(z|x) = \int q_\phi(z|a, x) q_\phi(a|x) \mathrm{d}a$. By making the inference network $q_\phi(z|a, x)$ depend on the stochastic variable $a$, the AVAE adds covariance structure to the posterior approximation over the stochastic unit $z$, since it no longer factorizes over its components $z^{(k)}$, i.e. $q_\phi(z|x) \neq \prod_k q_\phi(z^{(k)}|x)$. As discussed in the following, by factorizing the latent variables at each level of the hierarchy of BIVA we are able to achieve similar results without introducing additional

auxiliary variables in the model. To see this, we can focus for example on the highest latent variable $z_L$. In BIVA, the presence of the $z_i^{\text{BU}}$ variables makes the bottom-up inference path *stochastic*, as opposed to the deterministic BU path of the LVAE. While the conditional distribution $q_\phi(z_L|x, z_{<L}^{\text{BU}})$ still factorizes over the components of $z_L$, due to the stochastic BU variables the marginal distribution over $z_L$ no longer factorizes, i.e. $q_\phi(z_L|x) = \int q_\phi(z_L|x, z_{<L}^{\text{BU}})q_\phi(z_{<L}^{\text{BU}}|x)\mathrm{d}z_{<L}^{\text{BU}} \neq \prod_{k=1}^{K} q(z_L^{(k)}|x)$ . Therefore, the BU inference path enables the learning of a complex covariance structure in the higher TD stochastic latent variables, which is fundamental in the model to extract *good* high-level semantic features from the data distribution. Notice that, in BIVA, only $z_1^{\text{BU}}$ will have a marginally factorizing inference network.

### 3.3 Anomaly detection with BIVA

Anomaly detection is considered to be one of the most important applications of explicit density models. However, recent empirical results suggest that these models are not able to distinguish between two clearly distinctive data distributions [34], as they can assign a higher likelihood to data points from a data distribution that is very different from the one the model was trained on. Based on a thorough study, [34] states that the main issue is the fact that explicit density models tend to capture low-level statistics, as opposed to the high-level semantics that are preferable when doing anomaly detection. We hypothesize that the latent representations in the higher layers of BIVA can capture the high-level semantics of the data and that these can be used for improved anomaly detection.

In the standard ELBO from eq. (1), the main contribution to the expected log-likelihood term is coming from averaging over the variational distribution of the lower level latent variables. This will thus emphasize low-level statistics. So in order to perform anomaly detection with BIVA we instead need to emphasize the contribution from the higher layers. We can achieve this introducing an alternative score function inspired by the ELBO that partly replaces the inference network with the generative model, and uses therefore the generative hierarchy of the stochastic variables. In the following we define the hierarchy of stochastic latent variables as $\mathbf{z} = z_1, z_2, z_3, ..., z_L$ with $z_i = (z_i^{\text{BU}}, z_i^{\text{TD}})$. Instead of using as in the standard ELBO the variational approximation $q_\phi(\mathbf{z}|x)$ over all stochastic variables in the model, we use the prior distribution for the first $k$ layers and the variational approximation from the inference network for the others, i.e. $p_\theta(z_{\leq k}|z_{>k})q_\phi(z_{>k}|x, z_{\leq k}^{\text{BU}})$. In the computation of $q_\phi(z_{>k}|x, z_{\leq k}^{\text{BU}})$ we use samples $z_{\leq k}^{\text{BU}}$ from the inference network. Using this alternative distribution instead of $q_\phi(\mathbf{z}|x)$ in the ELBO in eq. (1), we define the score function for anomaly detection as:

$$\mathcal{L}^{>k} = \mathbb{E}_{p_\theta(z_{\leq k}|z_{>k})q_\phi(z_{>k}|x, z_{\leq k}^{\text{BU}})} \left[ \log \frac{p_\theta(x|\mathbf{z})p_\theta(z_{>k})}{q_\phi(z_{>k}|x, z_{\leq k}^{\text{BU}})} \right] \ . \tag{2}$$

$\mathcal{L}^{>0} = \mathcal{L}$ is the ELBO in eq. (1). As for the ELBO, we approximate the computation of $\mathcal{L}^{>k}$ with Monte Carlo integration. Sampling from $p_\theta(z_{\leq k}|z_{>k})q_\phi(z_{>k}|x, z_{\leq k}^{\text{BU}})$ can be easily performed by obtaining samples $\widehat{z}_{>k}$ from the inference network, that are then used to sample $\widehat{z}_{\leq k}$ from the conditional prior $p_\theta(z_{\leq k}|\widehat{z}_{>k})$.

$\mathcal{L}^{>k}$ with higher values of $k$ represents a useful metric for anomaly detection, as shown empirically in the experiments of Section 4.4. By only sampling the top $L - k$ variables from the variational approximation, in fact, we are forcing the model to only rely on the high-level semantics encoded in the highest variables of the hierarchy when evaluating this metric, and not on the low-level statistics encoded in the lower variables.

## 4 Experiments

BIVA is empirically evaluated by (i) an ablation study analyzing each novel component, (ii) likelihood and semi-supervised classification results on binary images, (iii) likelihood results on natural images, and (iv) an analysis of anomaly detection in complex data distributions. We employ a *free bits* strategy with $\lambda = 2$ [23] for all experiments to avoid latent variable collapse during the initial training epochs. Trained models are reported with 1 importance weighted sample, $\mathcal{L}_1$, and 1000 importance weighted samples, $\mathcal{L}_{1e3}$ [3]. We evaluate the natural image experiments by bits per dimension (bits/dim), $\mathcal{L}/(hwc \log(2))$, where $h$, $w$, $c$ denote the height, width, and channels respectively. For a detailed

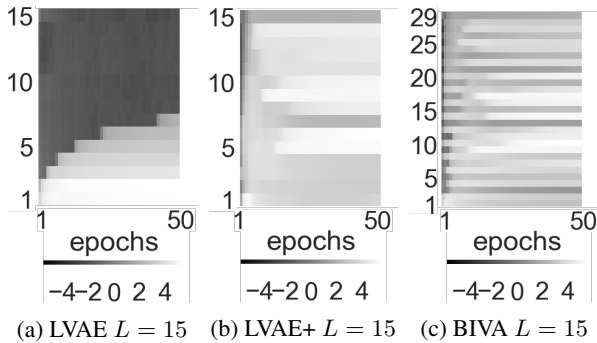

| (a) LVAE $L = 15$ | (b) LVAE+ $L = 15$ | (c) BIVA $L = 15$ |

Figure 2: The $\log KL(q||p)$ for each stochastic latent variable as a function of the training epochs on CIFAR-10. (a) is a $L = N = 15$ stochastic latent layer LVAE with no skip-connections and no bottom-up inference. (b) is a $L = N = 15$ LVAE+ with skip-connections and no bottom-up inference. (c) is a $L = 15$ stochastic latent layer ($N = 29$ latent variables) BIVA for which $1, 2, ..., N$ denotes the stochastic latent variables following the order $z_1^{\text{BU}}, z_1^{\text{TD}}, z_2^{\text{BU}}, z_2^{\text{TD}}, ..., z_L$.

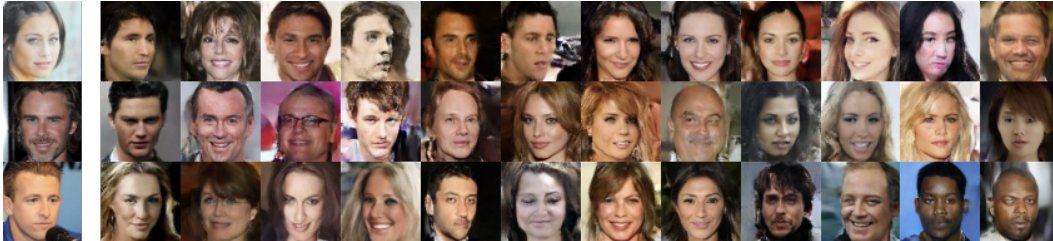

Figure 3: (left) images from the CelebA dataset preprocessed to 64x64 following [27]. (right) $\mathcal{N}(0, I)$ generations of BIVA with $L = 20$ layers that achieves a $\mathcal{L}_1 = 2.48$ bits/dim on the test set.

description of the experimental setup see Appendix C and the source code[1][2]. In Appendix D we test BIVA on complex 2d densities, while Appendix E presents initial results for the model on text.

## 4.1 Ablation Study

BIVA can be viewed as an extension of the LVAE from [50] where we add (i) extra dependencies in the generative model ($p_\theta(x|z_1) \to p_\theta(x|\mathbf{z})$ and $p_\theta(z_i|z_{i+1}) \to p_\theta(z_i|z_{>i})$) through the skip connections obtained with the deterministic top-down path and (ii) a bottom-up (BU) path of stochastic latent variables to the inference model. In order to evaluate the effects of each added component we define an LVAE with the exact same architecture as BIVA, but without the BU variables and the deterministic top-down path. Next, we define the LVAE+, where we add to the LVAE's generative model the deterministic top-down path. It is therefore the same model as in Figure 1 but without the BU variables. Finally, we investigate a LVAE+ model with $2L - 1$ stochastic layers. This corresponds to the depth of the hierarchy of the BIVA inference model $x \to z_1^{\text{BU}} \to \cdots \to z_{L-1}^{\text{BU}} \to z_L \to z_{L-1}^{\text{TD}} \to \cdots \to z_1^{\text{TD}}$. If this model is competitive with BIVA then it is an indication that it is the depth that determines the performance. The ablation study is conducted on the CIFAR-10 dataset against the best reported BIVA with $L = 15$ layers (Section 4.3), which means $2L - 1 = 29$ stochastic latent layers in the deep LVAE+.

Table 1 presents a comparison of the different model architectures. The positive effect of adding the skip connections in the generative models can be evaluated from the difference between the LVAE $L = 15$ and LVAE+ $L = 15$ results, for which there is close to a 0.2 bits/dim difference in the ELBO. Thanks to the more expressive posterior approximation obtained using its bidirectional inference network, BIVA improves the ELBO significantly w.r.t the LVAE+, by more than 0.3 bits/dim. Notice that a deeper hierarchy of stochastic latent variables in the LVAE+ will

Table 1: A comparison of the LVAE with no skip-connections and no bottom-up inference, the LVAE+ with skip-connections and no bottom-up inference, and BIVA. All models are trained on the CIFAR-10 dataset.

|  | PARAM. | BITS/DIM |
|---|---|---|
| LVAE L=15, $\mathcal{L}_1$ | 72.36M | $\leq 3.60$ |
| LVAE+ L=15, $\mathcal{L}_1$ | 73.35M | $\leq 3.41$ |
| LVAE+ L=29, $\mathcal{L}_1$ | 119.71M | $\leq 3.45$ |
| BIVA L=15, $\mathcal{L}_1$ | 102.95M | $\leq 3.12$ |

not necessarily provide a better likelihood performance, since the LVAE+ $L = 29$ performs worse than the LVAE+ $L = 15$ despite having significantly more parameters. In Figure 2 we plot for LVAE, LVAE+ and BIVA the KL divergence between the variational approximation over each latent variable

Table 2: Test log-likelihood on statically binarized MNIST for different number of importance weighted samples. The finetuned models are trained for an additional number of epochs with no *free bits*, $\lambda = 0$. For testing resiliency we trained 4 models and evaluated the standard deviations to be $\pm 0.031$ for $\mathcal{L}_1$.

| | $-\log p(x)$ |
|---|---|
| *With autoregressive components* | |
| PixelCNN [57] | $= 81.30$ |
| DRAW [13] | $< 80.97$ |
| IAFVAE [23] | $\leq 79.88$ |
| PixelVAE [14] | $\leq 79.66$ |
| PixelRNN [57] | $= 79.20$ |
| VLAE [5] | $\leq 79.03$ |
| *Without autoregressive components* | |
| Discrete VAE [42] | $\leq 81.01$ |
| | |
| **BIVA**, $\mathcal{L}_1$ | $\leq 81.20$ |
| **BIVA**, $\mathcal{L}_{1e3}$ | $\leq 78.67$ |
| **BIVA** finetuned, $\mathcal{L}_1$ | $\leq 80.47$ |
| **BIVA** finetuned, $\mathcal{L}_{1e3}$ | $\leq 78.59$ |

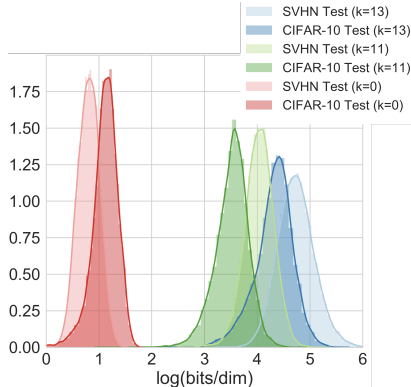

Figure 4: Histograms and kernel density estimation of the $\mathcal{L}^{>k}$ for $k = 13, 11, 0$ evaluated in bits/dim by a model trained on the CIFAR-10 train dataset and evaluated on the CIFAR-10 and the SVHN test set.

Table 3: Semi-supervised test error for BIVA on MNIST for 100 randomly chosen and evenly distributed labelled samples.

| | Error % |
|---|---|
| M1+M2 [22] | 3.33% ($\pm 0.14$) |
| VAT [32] | 2.12% |
| CatGAN [51] | 1.91% ($\pm 0.10$) |
| SDGM [31] | 1.32% ($\pm 0.07$) |
| LadderNet [38] | 1.06% ($\pm 0.37$) |
| ADGM [31] | 0.96% ($\pm 0.02$) |
| ImpGAN [44] | 0.93% ($\pm 0.07$) |
| TripleGAN [29] | 0.91% ($\pm 0.58$) |
| SSLGAN [6] | 0.80% ($\pm 0.10$) |
| | |
| **BIVA** | 0.83% ($\pm 0.02$) |

Table 4: Test log-likelihood on CIFAR-10 for different number of importance weighted samples. We evaluated two different BIVA with various number of layers ($L$). For testing resiliency we trained 3 models and evaluated the standard deviations to be $\pm 0.013$ for $\mathcal{L}_1$ and $L = 15$.

| | bits/dim |
|---|---|
| *With autoregressive components* | |
| ConvDRAW [12] | $< 3.58$ |
| IAFVAE $\mathcal{L}_1$ [23] | $\leq 3.15$ |
| IAFVAE $\mathcal{L}_{1e3}$ [23] | $\leq 3.12$ |
| GatedPixelCNN [56] | $= 3.03$ |
| PixelRNN [57] | $= 3.00$ |
| VLAE [5] | $\leq 2.95$ |
| PixelCNN++ [45] | $= 2.92$ |
| *Without autoregressive components* | |
| NICE [8] | $= 4.48$ |
| DeepGMMs [58] | $= 4.00$ |
| RealNVP [9] | $= 3.49$ |
| DiscreteVAE++ [54] | $\leq 3.38$ |
| Glow [21] | $= 3.35$ |
| Flow++ [16] | $= 3.08$ |
| | |
| **BIVA** L=10, $\mathcal{L}_1$ | $\leq 3.17$ |
| **BIVA** L=15, $\mathcal{L}_1$ | $\leq 3.12$ |
| **BIVA** L=15, $\mathcal{L}_{1e3}$ | $\leq 3.08$ |

and its prior distribution, $KL(q||p)$. This KL divergence is 0 when the two distributions match, in which case we say that the variable has collapsed, since its posterior approximation is not using any data-dependent information. We can see that while the LVAE is only able to utilize its lowest 7 stochastic variables, all variables in both LVAE+ and BIVA are active. We attribute this tendency to the deterministic top-down path that is present in both models, which creates skip-connections between all latent variables that allow to better propagate the information throughout the model.

## 4.2 Binary Images

We evaluate BIVA $L = 6$ in terms of test log-likelihood on statically binarized MNIST [43], dynamically binarized MNIST [28] and dynamically binarized OMNIGLOT [25]. The model parameterization and optimization parameters have been kept identical for all binary image experiments (see Appendix C). For each experiment on binary image datasets, we *finetune* each model by setting the free bits to $\lambda = 0$ until convergence in order to test the tightness of the $\mathcal{L}_1$ ELBO.

To the best of our knowledge, BIVA achieves state-of-the-art results on statically binarized MNIST, outperforming other latent variable models, autoregressive models, and flow-based models (see Table 2). Finetuning the model with $\lambda = 0$ improves the $\mathcal{L}_1$ ELBO significantly and achieves slightly better performance for the 1000 importance weighted samples. For dynamically binarized MNIST

|  | $\mathcal{L}^{>L-2}$ | $\mathcal{L}^{>L-4}$ | $\mathcal{L}^{>L-6}$ | $\mathcal{L}^{>0}$ |
|---|---|---|---|---|
| *Model trained on CIFAR-10:* | | | | |
| CIFAR-10 | 79.36 | 35.34 | 20.93 | 3.12 |
| SVHN | 121.04 | 58.82 | 26.76 | 2.28 |
| *Model trained on FashionMNIST:* | | | | |
| FASHIONMNIST | 228.38 | 107.07 | - | 94.05 |
| MNIST | 295.95 | 130.39 | - | 128.60 |

Table 5: The test $\mathcal{L}^{>k}$ for different values of $k$ and train/test dataset combinations evaluated in bits/dim for natural images and negative log-likelihood for binary images (lower is better).

and OMNIGLOT, BIVA achieves similar improvements with $\mathcal{L}_{1e3} = 78.41$ (state-of-the-art) and $\mathcal{L}_{1e3} = 91.34$ respectively, see Tables 10 and 11 in Appendix G.

**Semi-supervised learning.** BIVA can be easily extended for semi-supervised classification by adding a categorical variable $y$ to represent the class, as done in [22]. We add a classification model $q_\phi(y|x, z^{\mathrm{BU}}_{<L})$ to the inference network, and a class-conditional distribution $p_\theta(x|\mathbf{z}, y)$ to the generative model (see Appendix F for a detailed description). We train 5 different semi-supervised models on MNIST, each using a different set of just 100 randomly chosen and evenly distributed MNIST labels. Table 3 presents the classification results on the test set (mean and standard deviation over the 5 runs), that shows that BIVA achieves comparable performance to recent state-of-the-art results by generative adversarial networks.

### 4.3 Natural Images

We trained and evaluated BIVA $L = 15$ on 32x32 CIFAR-10, 32x32 ImageNet [57], and another BIVA $L = 20$ on 64x64 CelebA [27]. For the output decoding, we employ the discretized logistic mixture likelihood from [45] (see Appendix C for more details). In Table 4 we see that for the CIFAR-10 dataset BIVA outperforms other state-of-the-art non-autoregressive models and performs slightly worse than state-of-the-art autoregressive models. For the 32x32 ImageNet dataset BIVA achieves better performance than flow-based models, but the performance gap to the autoregressive models remains large (Table 13 in Appendix G). This may be due to the added complexity (more categories) of the 32x32 ImageNet dataset, requiring an even more flexible model. More research should be invested in defining an improved architecture for BIVA that holds more parameters and thereby achieves better performances.

Figure 3 shows generated samples from the $\mathcal{N}(0, I)$ prior of a BIVA $L = 20$ trained on the CelebA dataset. From a visual inspection, the samples are far superior to previous natural image generations by latent variable models. We believe that previous claims stating that this type of model can only generate *blurry* images should be disregarded [27]. Rather the limited expressiveness/flexibility of previous models should be blamed. Additional samples from BIVA can be found in Appendix G.

### 4.4 Does BIVA know what it doesn't know?

We test the anomaly detection capabilities of BIVA replicating the most challenging experiments of [34]. We train BIVA $L = 15$ on the CIFAR-10 dataset, and evaluate eq. (2) for various values of $k$ on the CIFAR-10 test set, the SVHN dataset [35] and the CelebA dataset. The results can be found in Table 5 and Figure 4, and are reported in terms of bits per dimension (lower is better). We see that for $k = 0$, corresponding to the standard ELBO, BIVA wrongly assigns lower values to data points from SVHN. This is in line with the results obtained with other explicit density models in [34], and shows that by using the standard ELBO the low-level image statistics prevail and the model is not able to correctly detect out-of-distribution samples. However, for higher values of $k$, the situation is reversed. We take this as an indication that BIVA uses the high-level semantics inferred from the data to better differentiate between the CIFAR-10 and the SVHN/CelebA distributions. We repeat the experiment training BIVA $L = 6$ on the FashionMNIST dataset (Table 5), and testing on the FashionMNIST test set and the MNIST dataset. Unlike the flow-based models used in [34], BIVA is able to learn a data distribution that can be used to detect anomalies with the standard ELBO (but also $k > 0$).

# 5 Conclusion

In this paper, we have introduced BIVA, that significantly improves performances over previously introduced probabilistic latent variable models and flow-based models. BIVA is able to generate natural images that are both sharp and coherent, to improve on semi-supervised classification benchmarks and, contrarily to other models, allows for anomaly detection using the extracted high-level semantics of the data.

## Footnotes

[1]Source code (Tensorflow): https://github.com/larsmaaloee/BIVA.

[2]Source code (PyTorch): https://github.com/vlievin/biva-pytorch.

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
