[Supplementary Material · BIVA_camera_ready_appendix_v2.pdf]

# (Appendix) BIVA: A Very Deep Hierarchy of Latent Variables for Generative Modeling

**Lars Maaløe**
Corti
Copenhagen
Denmark
lm@corti.ai

**Marco Fraccaro**
Unumed
Copenhagen
Denmark
mf@unumed.com

**Valentin Liévin & Ole Winther**
Technical University of Denmark
Copenhagen
Denmark
{valv,olwi}@dtu.dk

# A  Deep Learning and Variational Inference

The introduction of stochastic backpropagation [36, 18] and the variational auto-encoder (VAE) [24, 40] has made approximate Bayesian inference and probabilistic latent variable models applicable to machine learning problems considering complex data distributions, e.g. natural images, audio, and text. The VAE is a generative model parameterized by a neural network $\theta$ and is defined by an observed variable $x$ that depends on a hierarchy of stochastic latent variables $\mathbf{z} = z_1, ..., z_L$ so that: $p_\theta(x, \mathbf{z}) = p_\theta(x|z_1)p_\theta(z_L)\prod_{i=1}^{L-1} p_\theta(z_i|z_{i+1})$. This is illustrated in Figure 5a.

Figure 5: (a) Generative model of a VAE/LVAE with $L = 3$ stochastic variables, (b) VAE inference model, (c) LVAE inference model, and (d) skip connections among stochastic variables in the LVAE where dashed lines denote a skip-connection. Blue arrows indicate that there are shared parameters between the inference and generative model.

The distributions $p_\theta(z_i|z_{i+1})$ over the latent variables of the VAE are normally defined as Gaussians with diagonal covariance, whose parameters depend on the previous latent variable in the hierarchy (with the top latent variable $p_\theta(z_L) = \mathcal{N}(z_L; 0, I)$). The likelihood $p_\theta(x|z_1)$ is typically a Gaussian distribution for continuous data, or a Bernoulli distribution for binary data.

In order to learn the parameters $\theta$ we seek to maximize the log marginal likelihood over a training set: $\sum_i \log p_\theta(x_i) = \sum_i \log \int p_\theta(x_i, \mathbf{z}_i)d\mathbf{z}_i$. However, complex data distributions require an expressive model, which makes the above integral intractable. In order to circumvent this, we use Variational Inference [19] and introduce a posterior approximation $q_\phi(\mathbf{z}|x)$, known as *inference network* or *encoder*, that is parameterized by a neural network $\phi$. Using Jensen's inequality we can derive the *evidence lower bound* (ELBO), a lower bound to the integral in the marginal likelihood which is a function of the variational approximation $q_\phi(\mathbf{z}|x)$ and the generative model $p_\theta(x, \mathbf{z})$:

$$\log p_\theta(x) \geq \mathbb{E}_{q_\phi(\mathbf{z}|x)}\left[\log \frac{p_\theta(x, \mathbf{z})}{q_\phi(\mathbf{z}|x)}\right] \equiv \mathcal{L}(\theta, \phi) . \tag{3}$$

The parameters $\theta$ and $\phi$ can be optimized by maximizing the ELBO with stochastic backpropagation and the reparameterization trick, which allows using gradient ascent algorithms with low variance gradient estimators [24, 40]. As illustrated in Figure 5b, in a VAE the variational approximation is factorized with a bottom-up structure, $q_\phi(\mathbf{z}|x) = q_\phi(z_1|x)\prod_{i=1}^{L-1} q_\phi(z_{i+1}|z_i)$, so that each latent variable is conditioned on the variable below in the hierarchy. For ease of computation, all the factors in the variational approximation are typically assumed to be Gaussians whose mean and diagonal covariance are parameterized by neural networks.

**Latent variable collapse in VAEs.**  A deep hierarchy of latent stochastic variables will result in a more expressive model. However, the additional variables come at a price. As shown in [5, 30], we can rewrite the ELBO (eq. (1)):

$$\mathcal{L}(\theta, \phi) = \mathbb{E}_{q_\phi(\mathbf{z}|x)}\left[\log \frac{p_\theta(x, z_{<L}|z_L)}{q_\phi(z_{<L}|x)}\right] - \mathbb{E}_{q_\phi(z_{<L}|x)}\left[KL[q_\phi(z_L|z_{<L})||p_\theta(z_L))]\right] .$$

From the above, it becomes obvious that, during the optimization of the VAE, the top stochastic latent variables may have a tendency to *collapse* into the prior, i.e. $q_\phi(z_L|z_{<L}) = p_\theta(z_L) = \mathcal{N}(z_L; 0, I)$, if the model $p_\theta(x, z_{<L}|z_L)$ is powerful enough. This is supported by empirical results in [50, 2] amongst others. The tendency has limited the applicability of deep VAEs in problems with complex data distributions, and has pushed VAE research towards the extension of shallow VAEs with autoregressive models, that allow capturing a *lossy* representation in the latent space while achieving strong generative performances [14, 5]. Another research direction has focused on learning more complex prior distributions through normalizing flows [39, 52, 23]. Our research considers instead the original goal of building expressive models that can exploit a deeper hierarchy of stochastic latent variables while avoiding variable collapse.

|  |  |  |  |
|---|---|---|---|
| (a) Generative model | (b) BU inference | (c) TD inference | (d) Variable dependency |

Figure 6: A $L = 3$ layered BIVA with (a) the generative model, (b) bottom-up (BU) inference path, (c) top-down (TD) inference path, and (d) variable dependency of the generative models where dashed lines denote a skip-connection. Blue arrows indicate that the deterministic parameters are shared within the generative model or between the generative and inference model.

## B  Detailed Model Description

**Generative model.** The generative model (see Figure 6a) has a top-down path going from $z_L$ through the intermediary stochastic latent variables to $x$. Between each stochastic layer there is a ResNet block with $M$ layers set up similarly to [45]. Weight normalization [46] is applied in all neural network layers. In the generative model, the BU and TD units are not distinguished so we write $z_i = (z_i^{\text{BU}}, z_i^{\text{TD}})$. We use $f_{i,j}$ to denote the neural network function (a function of generative model parameters $\theta$) of ResNet layer $j$ associated with stochastic layer $i$. The feature maps are written as $d_{i,j}$. The generative process can then be iterated as $z_L \sim \mathcal{N}(0, I)$ and $i = L - 1, L - 2, \ldots, 1$:

$$d_{i,0} = z_{i+1} \tag{4}$$

$$d_{i,j} = < f_{\theta_{i,j}}(d_{i,j-1}); d_{i+1,j} > \textbf{ for } j = 1, ..., M \tag{5}$$

$$z_i = \mu_{\theta,i}(d_{i,M}) + \sigma_{\theta,i}(d_{i,M}) \otimes \epsilon_i , \tag{6}$$

where $d_{L,j} = 0$, $<;>$ denotes concatenation of feature maps in the convolutional network and hidden units in the fully connected network, $\epsilon \sim \mathcal{N}(0, I)$ and $\mu(\cdot)$ and $\sigma(\cdot)$ are parameterized by neural networks. To complete the generative model $p(x|\mathbf{z})$ is written in terms of $z_1$ and $d_1$ through a ResNet block $f_0$.

**Inference model.** The inference model (see Figure 6b and 6c) consists of a bottom-up (BU) and top-down (TD) paths such that bottom-up stochastic units only receive bottom-up information whereas the top-down units receive both bottom-up and top-down information. The top-down path shares parameters with the generative model. For each stochastic latent variable $z_i$ in $i = 1, ..., L$ we use a ResNet block with $M$ layers and there are associated neural network functions $g_{i,j}, j = 1, \ldots, M$ with parameters collectively denoted by $\phi$. The deterministic feature map of layer $i, j$ is denoted by $\tilde{d}_{i,j}$:

$$\tilde{d}_{i,0} = \begin{cases} x & i = 1 \\ < z_{i-1}; \tilde{d}_{i-1,M} > & \text{otherwise} \end{cases} \tag{7}$$

$$\tilde{d}_{i,j} = < g_{i,j}(\tilde{d}_{i,j-1}); \tilde{d}_{i-1,j} > \textbf{ for } j = 1, ..., M , \tag{8}$$

$$z_i^{\text{BU}} = \mu_i^{\text{BU}}(\tilde{d}_{i,M}) + \sigma_i^{\text{BU}}(\tilde{d}_{i,M}) \otimes \epsilon_i^{\text{BU}} \tag{9}$$

where $\epsilon \sim \mathcal{N}(0, I)$. Finally, to infer the top-down latent we use the bottom-up latent $z_i^{\text{TD}}$ inferred in eq. (9) and pass them through the generative path eq. (5) for $i = L - 1, L - 2, \ldots, 2$ to determine $d_{i,M}$ and

$$z_i^{\text{TD}} = \mu_i^{\text{TD}}(< \tilde{d}_{i,M}; d_{i,M} >) + \sigma_i^{\text{TD}}(< \tilde{d}_{i,M}; d_{i,M} >) \otimes \epsilon_i^{\text{TD}} . \tag{10}$$

## C   Experimental Setup

Throughout all experiments, we follow the BIVA model description that is described in detail in Appendix B and F.

**Optimization.**   All models are optimized using Adamax [20] with a hyperparameter setting similar to the one used in [23]. They are trained with a batch-size of 48 where the binary image experiments are trained on a single GPU and the natural image experiments are trained on two GPUs (by splitting the batch in 2 and then taking the mean over the gradients). For evaluation, we use exponential moving averages of the parameters space, similar to [23, 45].

**Binary image architecture.**   BIVA has $L = 6$ layers. The $g_{\phi_1}$ neural networks are defined by $M = 3$, 64x5x5 (number of kernels x kernel width x kernel height) convolutional layers and an overall stride of 2. Neural networks $i = 2, ..., 6$ are defined by four $M = 3$, 64x3x3 convolutional layers. The final neural network, $i = 6$, applies a stride of 2. All stochastic latent variables are densely connected layers of dimension $48, 40, 32, 24, 16, 8$ for $1, ..., L$ respectively. We apply a dropout rate of 0.5 for both the deterministic layers in the generative as well as the inference model.

**Natural image architecture (32x32).**   BIVA has $L = 15$ layers. The $g_{\phi_1}$ neural networks are defined by $M = 3$, 96x5x5 convolutional layers and an overall stride of 2. Neural networks $i = 2, ..., 15$ are defined by $M = 3$, 96x3x3 convolutional layers. Neural networks 11 and 15 are defined with a stride of 2. All stochastic latent variables are parameterized by convolutional layers with $38, 36, 34, ..., 10$ feature maps for $1, 2, 3, ..., L$ respectively. The kernel width and height of the stochastic latent variables are defined similarly to the dimension of the subsequent output after striding. We apply a dropout rate of 0.2 in the deterministic layers of the inference model.

**Natural image architecture (64x64).**   BIVA has $L = 20$ layers. The $g_{\phi_1}$ and $g_{\phi_2}$ neural networks are defined by $M = 3$, 64x7x7 and 64x5x5 convolutional layers respectively with a stride of 2 in each. Neural networks $i = 3, ..., 11$ are defined by $M = 3$ 64x3x3 convolutional layers. Neural network 11 is defined with a stride of 2. Neural networks $i = 12, ..., 20$ are defined by $M = 3$, 128x3x3 convolutional layers and network 20 has a stride of 2. All stochastic latent variables are parameterized by convolutional layers with $20, 19, 18, ..., 1$ feature maps for $1, 2, 3, ..., L$ respectively. The kernel width and height of the stochastic latent variables are defined similarly to the dimension of the subsequent output after striding. We apply a dropout rate of 0.2 in the deterministic layers of the inference model.

## D   Modeling Complex 2D Densities

| | POTENTIAL $U(\mathbf{Z})$ |
|---|---|
| **1:** | $\frac{1}{2}\left(\frac{\|\mathbf{z}\|-2}{0.4}\right)^2 - \ln\left(e^{-\frac{1}{2}\left[\frac{\mathbf{Z}_1-2}{0.6}\right]^2} + e^{-\frac{1}{2}\left[\frac{\mathbf{Z}_1+2}{0.6}\right]^2}\right)$ |
| **2:** | $\frac{1}{2}\left[\frac{\mathbf{Z}_2-w_1(\mathbf{Z})}{0.4}\right]^2$ |
| **3:** | $-\ln\left(e^{-\frac{1}{2}\left[\frac{\mathbf{Z}_2-\boldsymbol{w}_1(\mathbf{Z})}{0.35}\right]^2} + e^{-\frac{1}{2}\left[\frac{\mathbf{Z}_2-\boldsymbol{w}_1(\mathbf{Z})+\boldsymbol{w}_2(\mathbf{Z})}{0.35}\right]^2}\right)$ |
| **4:** | $-\ln\left(e^{-\frac{1}{2}\left[\frac{\mathbf{Z}_2-w_1(\mathbf{Z})}{0.4}\right]^2} + e^{-\frac{1}{2}\left[\frac{\mathbf{Z}_2-w_1(\mathbf{Z})+w_3(\mathbf{Z})}{0.35}\right]^2}\right)$ |
| | WITH $w_1(\mathbf{z}) = \sin\left(\frac{2\pi\mathbf{z}_1}{4}\right)$, $w_2(\mathbf{z}) = 3e^{-\frac{1}{2}\left[\frac{(\mathbf{Z}_1-1)}{0.6}\right]^2}$, $w_3(\mathbf{z}) = 3\sigma\left(\frac{\mathbf{Z}_1-1}{0.3}\right)$ AND $\sigma(x) = 1/\left(1+e^{-x}\right)$ . |

Table 6: Potentials defining the target densities $p(\mathbf{z}) = \frac{e^{-U(\mathbf{z})}}{Z}$.

**Problem.**   [31] showed that Variational Auto-Encoders can fit complex posterior distributions for the latent space using the inference model $q_\phi(z|x)$, parameterized as a fully factorized Gaussian and $p(x)$ being a simple diagonal Gaussian. In table 6, we define complex non-Gaussian densities using a potential model $U(\mathbf{Z})$, as described in [39]. While modeling such distributions remains

within the reach of an adequately complex Variational Autoencoder, optimizing such a model remains challenging.

**Objective.** Similarly to [31], we choose $p(x)$ to be an isotropic Gaussian and we model the target density using the top stochastic variable: $p(z_L) = \frac{e^{U(z)}}{Z}$. This results in the following bound:

$$\log Z \geq \mathbb{E}_{q_\phi(x,\mathbf{z})} \left[ U(z_L) + \log \frac{p_\theta(x|z_1)}{q_\phi(x)} + \sum_{i=1}^{L-1} \log \frac{p_\theta(z_i|z_{i+1})}{q_\phi(z_{i,TD}|z_{i+1},x)q_\phi(z_{i+1}|z_{i,BU},x)} \right] . \quad (11)$$

**Experimental Setup.** We test BIVA against the VAE and LVAE models using the same number of stochastic variables, hence the models use the same number of intermediate layers. All models are implemented using 5 stochastic layers, MLPs with one hidden layer of size 128 and with residual connections. The chosen architecture is voluntary kept minimal, therefore the task remains challenging for all models.

We train all models for $1e^4$ iterations using the Adamax optimizer. We use batch sizes of size 512. The potential is linearly annealed from 0.1 to 1 during $5e^3$ steps. In order to avoid posterior collapse, 0.5 *freebits* are applied to each stochastic layer. The learning rate is linearly increased from $1e^{-5}$ to $3e^{-3}$ and exponentially annealed back to $1e^{-5}$.

In order to measure the quality of the posterior density, we estimate $KL(q(z_L)||p(z_L))$ using $1e^6$ posterior samples evaluated using a grid of size $(-2, 2)^2$ with a resolution of $100 \times 100$. Each model is trained 100 times for each density.

**Results.** According to the approximate $KL(q(z_L)||p(z_L))$, we found that BIVA tends to learn a posterior that lies closer to the target density. Figure 7 shows that BIVA often learns more complex features than the baseline models, which posteriors remain closer to the modes. Figure 7 reveals that LVAE is able to find solutions that are competitive with the best BIVA samples according to $KL(q(z_L)||p(z_L))$. However, this happens very rarely whereas BIVA has a more robust optimization behaviour.

Figure 7: Distribution of the $KL(q(z_L)||p(z_L)))$ estimate for each model, each target density $p(z_L)$ and for different initial random seeds. We collected 100 runs for each model and for each density. We found that BIVA behaves more consistently and often yield better approximations than the baseline models.

# E  Initial Results on Text Generation Tasks

Optimizing generative models coupled with autoregressive models is a difficult task. Such coupling causes the posterior to collapse, and the latent variables are ignored. Nonetheless, autoregressive components remain a cornerstone of the generative models for text [2, 48, 49]. In order to enforce the model to use the latent variable, previous efforts aimed at weakening the decoder using powerful regularizing *tricks*, such as word dropout [2]. We investigate the use of BIVA in the context of sentence modeling without weakening the decoder. We show that it allows optimizing the latent variables more effectively, resulting in a higher measured KL when compared to the RNN-VAE [2] and the Hybrid VAE [48].

**Dataset.** We use the Bookcorpus dataset [60] of sentences of maximum 40 words, no preprocessing is performed and sentences are tokenized using the white spaces. We defined a vocabulary of 20000

Figure 8: Target densities $p(z_L)$ and the median posterior distributions $q(z_L)$ for each model according to $KL(q(z_L)||p(z_L)))$ out of 100 runs for each model and for each density.

| | PARAMETERS | $-\log p(x)$ | KL | PPL |
|---|---|---|---|---|
| *Results with autoregressive components, no dropout* | | | | |
| LSTM | $15.0M$ | $= 41.49$ | $-$ | 36.28 |
| RNN-VAE [2], $\mathcal{L}_1$, WARMUP | $23.7M$ | $\leq 42.09$ | 1.61 | 38.21 |
| RNN-VAE [2], $\mathcal{L}_1$, FINETUNED | $23.7M$ | $\leq 42.41$ | 5.13 | 39.26 |
| HYBRID VAE [48], $\mathcal{L}_1$, FINETUNED | $23.7M$ | $\leq 42.24$ | 4.67 | 38.70 |
| **BIVA** L=7, $\mathcal{L}_1$, FINETUNED | $23.0M$ | $\leq 42.34$ | 10.15 | 39.04 |
| *Results without autoregressive components, no dropout* | | | | |
| HYBRID VAE [48], $\mathcal{L}_1$, FINETUNED | $15.0M$ | $\leq 54.53$ | 14.10 | 112.1 |
| **BIVA** L=7 FINETUNED, $\mathcal{L}_1$ | $14.0M$ | $\leq 54.13$ | 15.33 | 108.3 |

Table 7: Test performances on the BookCorpus with 1 importance weighted sample (sentences limited to 40 words). The RNN-VAE and Hybrid VAE are are trained and evaluated from our own implementation.

words and filtered out the sentences that contain non-indexed tokens. We randomly sampled 10000 sentences for testing and used the remaining 56M sentences for training.

**Models.** We couple BIVA with an LSTM decoder, using the output of the convolutional model as an input sequence for the auto-regressive model. We compare our model against a LSTM language model [17], the RNN-VAE [2], and the Hybrid VAE [48], which couples a convolutional architecture with an LSTM decoder. We also perform experiments without using autoregressive components.

All LSTM models are parameterized by 1024 units and we use embeddings of dimension 512. This results in an RNN-VAE model with 23.7M parameters and we limit the other models to use the same total number of parameters. This results in using a limited number of stochastic layers for the BIVA and small a small number of kernels of 128.

**Training.** We trained the models for 5 epochs with an initial learning rate of $2e^{-3}$ using the Adamax optimizer. We used batches of size 512 and used only one stochastic sample. We train all latent variable models using the *freebits* method from [23] with an initial KL budget of 30 nats distributed equally over the stochastic variables and we incrementally decrease the *freebits* value *on plateau*. We also train the RNN-VAE baseline using the deterministic warmup method [2, 50] for comparison.

**Likelihood and latent variables usage.** We report the test set results in table 7 and test samples in 8 and reconstructions in table 9. While BIVA without the autoregressive decoder is not competitive with an LSTM language model, we observe that replacing the LSTM inference model by a BIVA model allows exploiting the latent space more actively, which results in a higher measured KL than the RNN-VAE and Hybrid VAE baselines.

| BIVA+LSTM | RNN-VAE |
|---|---|
| he said . | " two . |
| i tried to think of something to say to him , but he was already on his way back to the house . | " you do n't have to do this . " |
| it sounded as if he was going to say something . | the light from the lamp was dim , but the light was dim and the room was dark . |
| " and that 's why you 're coming . " | or a nuclear bomb , or something . |
| " what ? " | " the baby ? " |
| she swallowed . | " you 're not going to kill me . " |
| " i want you . " | she was n't going to . |
| glancing up , i saw the way he was staring at me with a look of pure hatred . | " i guess we could have been more careful , " he said . |
| i need a favor . " | there are some things that are not good . |
| he did n't . | " you 're a good man . |
| you 're not dead . | i had n't been able to get it out . |
| i stood , and he followed . | " you 're going to have to be careful , " he said . |
| " can i sit on the couch and talk ? " | it 's not a bad idea . |
| " it was n't until i was fifteen , i was n't in the mood to be around . | he asked . |
| i looked down at my lap . | " this is a bad idea , " he said , his voice a little hoarse . |
| the smile disappeared . | " i 'm sure he 's in love with you . |
| it was hard to tell which one was more of a rock . | as he stepped out of the car , he saw the man standing in the doorway , his eyes wide and his face pale . |
| i 'm not sure it 's a good idea . | . |
| the first two . | " no . |
| he was there . | " in the meantime , i need to get some sleep , " i said . |
| " all of you , " joe said . | i was n't . |
| he did n't care if he was n't a vampire . | did i want to talk to you ? |
| her mouth curved up , then she nodded . | " i want to hear you say it . " |
| just tell me what you want in the end . | the train was already in the driveway . |
| and again . | " good . |
| the other man 's voice was hoarse and ragged . | i just needed to get out of here , and i needed to get out of here . |
| i had n't known that was a bad idea , but i had n't been able to get it out of my head . | " this is a good idea . |
| your brother is the most important thing to me . | " hey . " |
| you dont need to go to the police , right ? | she took a deep breath and let it out . |
| there was a long silence . | then he kissed her . |
| i looked up . | i felt a warm hand on my shoulder and a warm smile spread across my face . |
| he nodded , and he looked at me , and i could tell he was thinking about it . | " he 's dead . " |
| " hang on , baby . | at the time , i was going to have to get out of the house . |
| we had to be close to the city , and we could n't afford to be here . | he was so close to the edge of the bed . |
| you know , it would be better if you were n't so stupid . " | " i do n't know . |
| excuse me ? | " i do n't have a choice . " |
| you know how much i love you , too . | i know i 'm not going to let him touch me , but i do . |
| a woman 's voice , a voice that was familiar . | i could n't see the face of the man who 'd just been in the doorway . |
| i have a very important business to attend to , and i 'm going to have to make a decision . | in the end , we all know that we are not going to be able to get out of this . |
| they sat on the small wooden table in the center of the room . | " yes . |
| " it 's fine . " | " what are you doing here ? " |
| she felt a rush of relief . | so the only thing that mattered was that he was here . |
| maria , he says . | neither of them spoke . |
| what ? | from now on , you will be able to get out of here . |
| " it does n't seem like a lot to me , " he said . | the thought of having to kill him made him want to kill her . |
| he 'd told her everything . | the other two were staring at me , their eyes wide . |
| " she 's in shock . | i did n't want to be a part of it , but i was n't going to let it go . |
| " after all , " he murmured , " i 'm going to go get the rest of the stuff . " | " i do n't want to talk about it . |
| and then , finally , she 'd done it . | she looked at him , her eyes wide . |
| her words were a whisper , but it was n't enough . | " that 's what you 're going to do . |

Table 8: Samples decoded from the prior of the BIVA with LSTM decoder and baseline RNN-VAE.

| input | BIVA+LSTM | RNN-VAE |
|---|---|---|
| " a sad song , being sung alone in the basement . " | " it sounds like you 've been through a lot . " | " you 're going to be a great father . " |
| more often , though , wherever she sank , beck was there . | more than anything , she wanted to be with him . | in the end , we all knew what was going on . |
| he looked just about as pale as i had ever seen him . | he 's still a lot more than a friend . | he was n't going to let her go . |
| caleb turned and shoved him back as he took his true form . | he lifted me up , his arms still wrapped around my waist . | he was standing in the doorway , his hands folded in front of him . |
| i gasped , tried to pull away , squeezed my legs together . | i gasped , and he was n't able to stop himself . | i felt my body tense , and i could n't help but smile . |
| i agreed as i adjusted myself and sat heavily in my chair . | i tried to ignore it , but my eyes were still closed . | i did n't want to be the one to tell him . |
| you bind me , UNK in darkness , though , in light . | he 'd decided to take her home , to make her feel safe . | he was more than willing to let her go . |
| they promise me things , ask me questions , whisper and plead . | they might be able to do something about it , but they do n't . | " we need to talk , " he said , his voice low . |
| i glowed as i held the bear , almost bigger than me . | i started to close my eyes , but he was too strong . | i could n't help but smile at the sight of her . |
| i wonder how much he pays them to be his guard dogs . | i had to admit that it was n't a good idea . | i do n't want to be a part of this . |
| " hmmm , " richard muttered , and headed up the path . | " jesus , " he said , his voice barely audible . | " but you 're going to be a father . |
| he was happy that he had found it in the UNK hall . | he was n't going to be the one to go . | he was n't sure if he was going to make it . |
| at the shack , at the condo , at the hangar . " | at the moment , the only thing that mattered was that he was n't alone . | he was staring at the floor , his eyes wide . |
| " i 'd pop to go to the dance with you . " | " i 'd prefer to go to the hospital . | " i 'm going to go to the bathroom . |
| someday , i 'll share them with the rest of the world . | and now i have a lot of my own . | " we 're going to have to do something about it . |
| " maybe i 'm not the right person for this one " . | " maybe we can get a little more of a ride . " | " i do n't think you 're going to be able to do that . " |
| " gin is my sister , and she 's coming with me . | " there 's a chance i can get a little more sleep . " | " if you want to , i 'll be there . " |
| thick desire stormed her ... along with a bittersweet curl of emotion . | the tension was gone , and he was n't looking at me . | the air smelled of stale cigarette smoke . |
| they caused him to stagger back and drove him to the ground . | they had to be at the top of the hill . | he 'd found a way to get her to safety . |
| you 're not much of a friar , friar , he says . | you 're not supposed to be around here , are you ? " | you 're not going to be able to do that , are you ? " |

Table 9: Reconstruction of samples from the test set using the BIVA with LSTM decoder and the RNN-VAE baseline. The samples are decoded from the posterior distribution by using greedy decoding, without teacher forcing.

# F Semi-Supervised Learning

When defining BIVA for semi-supervised classification tasks we follow the approach described for the M2 model in [22]. In addition to BIVA, described in detail in Appendix B, we introduce a classification model $q_\phi(y|x, z^{\text{BU}}_{<L})$ in the inference model, where $y$ is the class variable, and a Categorical latent variable dependency in the generative model.

**Inference model.** For the classification model we introduce another deterministic hierarchy with an equivalent parameterization as $\tilde{d}_{i,1}, ..., \tilde{d}_{i,M}$. We denote the hierarchy $\tilde{d}^{\text{c}}_{i,1}, ..., \tilde{d}^{\text{c}}_{i,M}$. The forward-pass is performed by:

$$\tilde{d}^{\text{C}}_{i,0} = \begin{cases} x & i = 1 \\ \tilde{d}^{\text{c}}_{i-1,M} & \text{otherwise} \end{cases} \tag{12}$$

$$\tilde{d}^{\text{C}}_{i,j} = < g^{\text{C}}_{\phi_{i,j}}(\tilde{d}^{\text{c}}_{i,j-1}); z^{\text{BU}}_i > \quad \textbf{for } j = 1, ..., M \tag{13}$$

$$y = g^{\text{c}}_{\phi_{i,M+1}}(\tilde{d}^{\tilde{\text{c}}}_{i,M}) , \tag{14}$$

where $g^{\text{c}}_{\phi_{i,M+1}}$ is a final densely connected neural network layer, of the same dimension as the number of categories, and a Softmax activation function. The inference model is thereby factorized by:

$$q_\phi(\mathbf{z}, y|x) = q_\phi(z_L|x, y, z^{\text{BU}}_{<L})q_\phi(y|x, z^{\text{BU}}_{<L}) \prod_{i=1}^{L-1} q_\phi(z^{\text{BU}}_i|x, z^{\text{BU}}_{<i})q_{\phi,\theta}(z^{\text{TD}}_i|x, y, z^{\text{BU}}_{<i}, z^{\text{BU}}_{>i}, z^{\text{TD}}_{>i}) . \tag{15}$$

**Generative model.** For each stochastic latent variable, $\mathbf{z}$, and the observed variable $x$ in the generative model, as well as the TD path of the inference model, we add a conditional dependency on a categorical variable $y$:

$$p_\theta(x, y, \mathbf{z}) = p_\theta(x|\mathbf{z}, y)p_\theta(z_L)p_\theta(y) \prod_{i=1}^{L-1} p_\theta(z_i|z_{>i}, y) . \tag{16}$$

**Evidence lower bound.** In a semi-supervised learning problem, we have labeled data and unlabeled data which results in two formulations of the ELBO. The ELBO for labeled data points is given by:

$$\log p_\theta(x, y) \geq \mathbb{E}_{q_\phi(\mathbf{z}|x,y))} \left[ \log \frac{p_\theta(x, y, \mathbf{z})}{q_{\phi,\theta}(\mathbf{z}|x, y)} \right] \equiv -\mathcal{F}(\theta, \phi) . \tag{17}$$

Since the classification model is not included in the above definition of the ELBO we add a classification loss term (a categorical cross-entropy), equivalent to the approach in [22]:

$$\bar{\mathcal{F}}(\theta, \phi) = \bar{\mathcal{F}}(\theta, \phi) - \alpha \cdot \mathbb{E}_{q(z<L|x)}[\log q_\phi(y|x, z^{\text{BU}}_{<L})] , \tag{18}$$

where $\alpha$ is a hyperparameter that we define as in [31]. For the unlabeled data points, we marginalize over the labels:

$$\log p_\theta(x) \geq \mathbb{E}_{q_\phi(\mathbf{z}, y|x)} \left[ \log \frac{p_\theta(x, y, \mathbf{z})}{q_{\phi,\theta}(\mathbf{z}, y|x)} \right] \equiv -\mathcal{U}(\theta, \phi) . \tag{19}$$

The combined objective function over the labeled, $(x_l, y_l)$, and unlabeled data points, $(x_u)$, are thereby given by:

$$\mathcal{J}(\theta, \phi) = \sum_{x_l, y_l} \bar{\mathcal{F}}(\theta, \phi; x_l, y_l) + \sum_{x_u} \mathcal{U}(\theta, \phi; x_u) . \tag{20}$$

# G Additional Results

Table 10: Test log-likelihood on dynamically binarized MNIST for different number of importance weighted samples. The finetuned models are trained for an additional number of epochs with no *free bits*, $\lambda = 0$.

|  | $-\log p(x)$ |
|---|---|
| *Results with autoregressive components* | |
| DRAW+VGP [53] | $< 79.88$ |
| IAFVAE [23] | $\leq 79.10$ |
| VLAE [5] | $\leq 78.53$ |
| *Results without autoregressive components* | |
| IWAE [4] | $\leq 82.90$ |
| CONVVAE+HVI [47] | $\leq 81.94$ |
| LVAE [50] | $\leq 81.74$ |
| DISCRETE VAE [42] | $\leq 80.04$ |
| **BIVA**, $\mathcal{L}_1$ | $\leq 80.60$ |
| **BIVA**, $\mathcal{L}_1 e3$ | $\leq 78.49$ |
| **BIVA** FINETUNED, $\mathcal{L}_1$ | $\leq 80.06$ |
| **BIVA** FINETUNED, $\mathcal{L}_{1e3}$ | $\leq 78.41$ |

Table 11: Test log-likelihood on dynamically binarized OMNIGLOT for different number of importance weighted samples. The finetuned models are trained for an additional number of epochs with no *free bits*, $\lambda = 0$.

|  | $-\log p(x)$ |
|---|---|
| *Results with autoregressive components* | |
| DRAW [13] | $< 96.50$ |
| CONVDRAW [12] | $< 91.00$ |
| VLAE [5] | $\leq 89.83$ |
| *Results without autoregressive components* | |
| IWAE [4] | $\leq 103.38$ |
| LVAE [50] | $\leq 102.11$ |
| DVAE [42] | $\leq 97.43$ |
| **BIVA**, $\mathcal{L}_1$ | $\leq 95.90$ |
| **BIVA** FINETUNED, $\mathcal{L}_1$ | $\leq 93.54$ |
| **BIVA** FINETUNED, $\mathcal{L}_{1e3}$ | $\leq 91.34$ |

Table 12: Test log-likelihood on statically binarized Fashion MNIST for different number of importance weighted samples. The finetuned models are trained for an additional number of epochs with no *free bits*, $\lambda = 0$.

|  | $-\log p(x)$ |
|---|---|
| **BIVA**, $\mathcal{L}_1$ | $\leq 94.05$ |
| **BIVA** FINETUNED, $\mathcal{L}_1$ | $\leq 93.54$ |
| **BIVA** FINETUNED, $\mathcal{L}_{1e3}$ | $\leq 87.98$ |

Table 13: Test log-likelihood on ImageNet 32x32 for different number of importance weighted samples.

|  | BITS/DIM |
| --- | --- |
| *With autoregressive components* | |
| CONVDRAW [12] | $< 4.10$ |
| PIXELRNN [57] | $= 3.63$ |
| GATEDPIXELCNN [56] | $= 3.57$ |
| *Without autoregressive components* | |
| REALNVP [9] | $= 4.28$ |
| GLOW [21] | $= 4.09$ |
| FLOW++ [16] | $= 3.86$ |
| | |
| **BIVA**, $\mathcal{L}_1$ | $\leq 3.98$ |
| **BIVA**, $\mathcal{L}_{1e3}$ | $\leq 3.96$ |

(a) $\mathcal{L}_1$ (bits/dim).

(b) $\log p_\theta(x|\mathbf{z})$ (bits/dim).

Figure 9: Convergence plot on CIFAR-10 training for the LVAE with $L = 15$, the LVAE+ with $L = 15$, the LVAE+ with $L = 29$, and BIVA with $L = 15$. (a) shows the convergence of the 1 importance weighted ELBO, $\mathcal{L}_1$, calculated in bits/dim. (b) shows the convergence of the *reconstruction loss*. The discrepancy between (a) and (b) is explained by the added cost from the stochastic latent variables, the Kullback-Leibler divergence $KL[p(\mathbf{z})||q(\mathbf{z}|x)]$.

Figure 10: 64x64 CelebA samples generated from a BIVA with increasing levels of stochasticity in the model (going from close to the mode to the full distribution). In each column the latent variances are scaled with factors $0.1, 0.3, 0.5, 0.7, 0.9, 1.0$. Images in a row look similar because they use the same Gaussian random noise $\epsilon$ to generate the latent variables. BIVA has $L = 20$ stochastic latent layers connected by three layer ResNet blocks.

(a) $\sigma^2 = 0.01$

(b) $\sigma^2 = 0.1$

(c) $\sigma^2 = 0.5$

(d) $\sigma^2 = 1.0$

Figure 11: BIVA $\mathcal{N}(0, \sigma^2)$ generations with varying $\sigma^2 = 0.01, 0.1, 0.5, 1.0$ for (a), (b), (c) and (d) respectively. We follow the same generating procedure of Figure 10. BIVA has $L = 20$ stochastic latent variables and is trained on the CelebA dataset, preprocessed to 64x64 images following [27]. BIVA achieves a $\mathcal{L}_1 = 2.48$ bits/dim on the test set. Close to the mode of the latent distribution there is very little variance in generated natural images. When we *loosen* the samples towards the full distribution, $\sigma^2 = 1$, we can see how the generated images are adopting different styles and contexts.

Figure 12: BIVA $L = 20$ generations (right) from fixed $z_{>i}$ given an input image (left), for different layers throughout the stochastic variable hierarchy (from left to right $i = 12, 14, 16, 17, 18, 19$). The model is trained on CelebA, preprocessed to 64x64 images following [27]. $z_{>i}$ are fixed by passing the original image through the encoder, after which $z_{\leq i}$ are sampled from the prior. When generating from a higher $z_i$ (columns) it is shown how the model has more *freedom* to augment the input images. BIVA achieves a $\mathcal{L}_1 = 2.48$ bits/dim on the test set.

Figure 13: BIVA $\mathcal{N}(0, I)$ generations on a model trained on CIFAR-10. BIVA has $L = 15$ stochastic latent variables and achieves a 3.08 bits/dim on the test set. The images are still not as sharp and coherent as the PicelCNN++ [45] (3.08 vs. 2.92), however, it does achieve to find coherent structure resembling the categories of the CIFAR-10 dataset.