[Reviews · NeurIPS 2019]

Reviewer 1



Originality: the techniques used in the methods seem to be incremental but it is exciting to see that a very deep latent variable model can solve some problems that the shallow ones do not. Contribution&significance: the method is evaluated on a wide range of tasks and the results are excellent both quantitatively and qualitatively, compared with a large family of deep generative models. However, I worry about the reproducibility since most of the results are run by only once. Clarity: the organization and writing of the paper are good and the related work is clearly discussed.

Reviewer 2



This is a good paper in general. It is well written and easy to follow. The motivation and insight is clear, the proposed method is reasonable and novel, and the experimental results are convince. I have several small questions: For the equation between line 135 and 136( why does it not have a equation number?): It says that z^{TD} is conditioned on z^{BU}_{>i}. However, it seems z^{TD} should only depends on the BU variables lower than it. Is it a typo? The experiments stops on L=20. What is the performance if we keep increasing L to 30, 40, or 100, 200, 1000? While the results shown in table 5 provides some insight, it would be good to understand more about the learned hierarchical representation. For example, what do z^1, z^2, .... z^L represent, respectively?

Reviewer 3



The authors propose variational autoencoder where inference is performed bidirectionally (top -> bottom and bottom -> top) with the intention of enhance the flow of information and avoid inactive units. This is achieved via multi-layered stochastic variables and a deterministic backbone network. The proposed inference model is akin to ladder VAE but with stochastic layers. The proposed model does not contain autorregressive elements. The authors present extensive results on image datasets and also consider semisupervised classification and outlier detection tasks. It is not clear why p_\theta(x|z) is conditioned on the entire set of latent variables rather than z_1 as in Figure 1a, unless off course this is the case when accounting for the (deterministic network) skip connections in Figure 1d. In either case, it needs to be clarified. Similarly, why is q(z_i^BU|x,z_i^BU,z_>i^TD) dependent on x considering Figure 1c? From the description in Section 3.2 it seems clear that the model does not explicitly use skip connections as in Figure 1d but the deterministic path implicitly acts as skip connection (and conditioning on x) serving as backbone for all the layers. That being said, Figure 1 does not seem to help explain the concept behind the model. I enjoyed reading the paper (minus the graphical model), it is well written, well motivated and the experiments are well thought, extensive and convincing. Post-rebuttal: The changes to the model description and graphical model, as well as the error bars on the results are welcome additions to the revision.

[Author Response · NeurIPS 2019]

Thank you very much for the thorough and generally positive feedback. The following response should act as a comprehensive response to all comments given.

**R1.1** *However, I worry about the reproducibility since most of the results are run by only once.*

**A1.1** Upon acceptance we will publish the source code, implemented in Tensorflow, that was also submitted with this paper for review. In the meantime we have developed an implementation in PyTorch that will also be released upon acceptance. We have run the MNIST experiments 4 times and CIFAR-10 experiments 3 times, with the provided code, and the results are added in the table below, which shows reproducibility - we will add confidence intervals in the final paper. Across experiments we used equivalent hyperparameters, also across datasets, which should alleviate potential reproducibility issues.

| STAT. BIN. MNIST | $-\log p(x)$ | CIFAR-10 | BITS/DIM |
|---|---|---|---|
| ... | ... | ... | ... |
| **BIVA**, $\mathcal{L}_1$ | $\leq 81.20 \pm 0.031$ | **BIVA** L=15, $\mathcal{L}_1$ | $\leq 3.13 \pm 0.013$ |
| ... | ... | ... | ... |

**R2.1** *For the equation between line 135 and 136( why does it not have a equation number?): It says that $z^{TD}$ is conditioned on $z^{BU}_{>i}$. However, it seems $z^{TD}$ should only depends on the BU variables lower than it. Is it a typo?*

**A2.1** We will add an equation number. We have taken the comments from **Reviewer 3** into account and have redone the graphical model to make it easier to read. The $z_i^{TD}$ stochastic layers are indeed dependent on the bottom-up stochastic layers above and below.

**R2.2** *The experiments stops on L=20. What is the performance if we keep increasing L to 30, 40, or 100, 200, 1000?*

**A2.2** From the experiments, we saw that for a 32x32 image multiple stochastic layers started to be inactive for $L > 15$ and for 64x64 images this was not the case, however, we ran out of GPU memory for $L > 20$.

**R2.3** *While the results shown in table 5 provides some insight, it would be good to understand more about the learned hierarchical representation. For example, what do $z^1$, $z^2$, .... $z^L$ represent, respectively?*

**A2.3** We agree that a thorough investigation of variables would make for an interesting supplement to this research. In the Appendix (figure 11) the effect of the different layers on the generation of CelebA images is visualized. In the experiment, we generated the same image multiple times while fixing different sets of variables, from highest to lowest. It is shown how the model changes attributes (e.g. glasses) when increasing the number of stochastic variables that we sample from.

**R3.1** *It is not clear why $p_\theta(x|z)$ is conditioned on the entire set of latent variables rather than $z_1$ as in Figure 1a, unless off course this is the case when accounting for the (deterministic network) skip connections in Figure 1d. In either case, it needs to be clarified.*

**A3.1** You are correct, when writing $p_\theta(x|\mathbf{z})$ we are taking into account the dependencies from the deterministic network (skip connections in Figure 1d). We will clarify this in the main text.

**R3.2** *That being said, Figure 1 does not seem to help explain the concept behind the model.*

**A3.2** Thanks for the feedback, we agree that Figure 1 can be confusing. We will simplify the figure as shown below, only highlighting the variable dependency in the graphical model and not the deterministic nodes. The figure with the deterministic nodes will be clarified and moved to the appendix to help people interested in implementing the model.

(a) Graphical model of the generative model

(b) Graphical model of the variational approximation

[Meta-Review · NeurIPS 2019]

This paper was reviewed by three expert reviewers and received two Accept and one Clear Accept recommendations. All the three reviewers are positive about this paper, and agree that this paper is well written and well motivated. The proposed method is reasonable and novel, and experiments are convincing. The rebuttal further addresses reviewers' comments. Therefore, the AC recommends accepting the paper.